# Multicomponent Training and Optimal Dosing Strategies for Adults with Hypertension: A Systematic Review and Meta-Analysis of Randomized Controlled Trials

**DOI:** 10.3390/sports11060115

**Published:** 2023-06-08

**Authors:** Isabel López-Ruiz, Fernando Lozano, María Dolores Masia, Noelia González-Gálvez

**Affiliations:** 1Facultad del Deporte UCAM, Universidad Católica de Murcia, 30107 Murcia, Spain; imlopez7@alu.ucam.edu; 2General University Hospital of Ciudad Real, 13005 Ciudad Real, Spain; drlozano68@gmail.com; 3Faculty of Health Sciences, University Hospital San Juan de Alicante, 03550 Alicante, Spain; mariadomasia@hotmail.com

**Keywords:** blood pressure, exercise, physical activity, cardiovascular disease

## Abstract

(1) Background: Non-pharmacological interventions have demonstrated efficacy in the prevention, management, and control of hypertension. Multicomponent training confers a host of benefits to the general populace. The aim of this research was to assess the impact of multicomponent training on the blood pressure of adults with hypertension and ascertain the nature of the dose–response relationship. (2) Methods: This systematic review adhered to the PRISMA guidelines and was registered in PROSPERO. Eight studies were included, following a literature search across PubMed, Web of Science, Cochrane, and EBSCO. Randomized controlled trials implementing multicomponent training interventions on adults with hypertension were considered for inclusion. A quality assessment was performed using the PEDro scale, with a random-effects model utilized for all analyses. (3) Results: Multicomponent training yielded a significant reduction in systolic (MD = −10.40, *p* < 0.001) and diastolic (MD = −5.97, *p* < 0.001) blood pressure relative to the control group. Interventions lasting over 14 weeks with a minimum frequency of three sessions per week, each lasting 60 min, were deemed most effective. (4) Conclusion: An optimal training intensity was achieved with 30 min of aerobic exercise at 75% of the heart rate reserve, whereas sets of 10 repetitions at 75% of one repetition maximum produced the best outcomes in strength training.

## 1. Introduction

Cardiovascular disease is the leading cause of premature death worldwide, and hypertension is one of its most prevalent risk factors [1,2]. Approximately one-third of the world’s adult population suffers from hypertension, a factor that increases the risks of heart disease, encephalopathy, and kidney disease, with hypertension topping the list of serious, non-communicable diseases responsible for 10.4 million deaths per year [3]. 

Non-pharmacological interventions have been shown to be effective in the prevention, management, and control of hypertension [4]. The World Health Organization recommend that all adults should undertake at least 150 min of moderate-intensity activity or at least 75 min of vigorous-intensity physical activity, or some equivalent combination of moderate-intensity and vigorous-intensity aerobic physical activity, per week. In addition, the guidelines recommend regular muscle-strengthening activity for all age groups [5]. In this regard, the effect of cardiovascular training on blood pressure levels has been well established in the literature, with reductions ranging from −4.9 to −12 mm Hg for systolic blood pressure (SBP) and from −3.4 to −5.8 mm Hg for diastolic blood pressure (DBP) [6]. Strength training has shown improvements in blood pressure ranging from −3.0 to −4.7 mm Hg for SBP and from −3.2 to −3.8 mm Hg for DBP [6]. 

A description of the combined effect of both cardiovascular and strength training has been attempted. Multicomponent training has been shown to be more effective than aerobic training for improving some variables [7] and has been widely used and recommended in the older adult population to reduce frailty [8]. In multicomponent training, work is performed with many types of exercise by combining the physical capacities of strength, cardiovascular endurance, flexibility, and balance in the same session [9]. This training has shown positive effects on the health of the adults, with haemodynamic improvements and improvements in arterial stiffness observed [10], as recommended by the American College of Sports Medicine and by the American Heart Association [11]. 

Three similar systematic reviews have been conducted thus far. However, some aspects should be taken into consideration. The study by Corso et al. [12] included subjects with and without cardiovascular pathologies, such as diabetes, as well as subjects with hypertension; it also included training in a single session (multicomponent training) and in several sessions (concurrent training). The statistical analysis with meta-analyses by Herrod et al. [13] focused on the older adult population, and Pescatello et al. [14] did not include meta-analyses.

On the other hand, many differences have been observed between the dosage of the training protocols applied (time, frequency, duration, intensity, or order of components) [15,16,17,18,19,20,21,22,23], and a good understanding of the optimal dosage of the multicomponent training according to the blood pressure levels of patients with hypertension would help optimise the prescription, control, and management of hypertension. 

The present systematic review and meta-analysis hypothesises that (a) multicomponent training will be effective in reducing systolic and diastolic hypertension levels in subjects with hypertension, and (b) the multicomponent training protocol that will show the greatest benefits for blood pressure will apply moderate to vigorous intensities in cardiovascular work and moderate to high intensities in strength training. In addition, the present systematic review with a meta-analysis aims (a) to evaluate the effects of multicomponent training on SBP and DBP levels of adults with hypertension, and (b) to identify the optimal dosage of multicomponent training to manage hypertension in adults.

## 2. Materials and Methods

### 2.1. Study Design

This systematic review and meta-analysis adhered to the Preferred Reporting Items for Systematic Reviews and Meta-Analyses (PRISMA) guidelines [24]. In addition, this research followed the Cochrane Handbook for Systematic Reviews of Interventions [25] and was registered in PROSPERO (number CRD42021247395).

### 2.2. Eligibility Criteria

The inclusion criteria for the articles were (a) a randomised controlled trial (RCT) design, (b) the study of adults (30–80 years old) with hypertension, (c) a multicomponent training intervention, (d) the inclusion of a control group that did not engage in any form of exercise, (e) 100% supervision, and (f) articles written in English, Spanish, or Portuguese.

The exclusion criteria were (a) multicomponent training that did not include physical exercise, (b) a sample composed of pregnant women and/or patients with hypertensive and with severe diseases that precluded safe physical exercise (unstable coronary artery disease, heart or kidney failure, severe pulmonary hypertension, or uncontrolled diabetes), and (c) short communications, notes, letters, review articles, or brief reports.

### 2.3. Search Strategy

Four electronic databases (PubMed, Cochrane, WOS, and EBSCO) were used for the search, which ended in April 2022. No limitations for the starting year were defined in the search. The following search terms were used: hypertension, blood pressure, systolic, diastolic, and hypertensive, which were combined using the Boolean AND operator, with the following: exercise, training, sports, physical*, rehabilitat*, fit*, train*, strength*, aerobic*, endurance*, weight*, HIIT, MICT, fitness, resistance, combined, and multicomponent. The search strategy used in each database is detailed in Appendix A. A total of nine studies were included in the study (Figure 1).

### 2.4. Data Collection and Synthesis

Two reviewers (I.L.R. and N.G.G.) independently screened the literature in the selected databases using the search terms, considering the inclusion and exclusion criteria. In the case of any discrepancy regarding the inclusion of a given study, the data extraction or assessment were repeated without looking at the reviewer’s previous information.

### 2.5. Data Extraction and Study Quality

A Physiotherapy Evidence Database (PEDro) score was used to assess the individual study of the quality. The score obtained showed a strong validity and inter-rater reliability for the assessment of RCTs [26,27,28]. A risk of bias summary graph was created to identify the authors’ judgments, broken down according to each risk of bias criterion in all included studies.

Data extraction (Table 1) and a quality assessment were performed by two reviewers (I.L.R. and N.G.G.) independently. Disagreements were resolved by repeating the data extraction or assessment without looking at the information previously reported by the reviewer. To determine the inter-reviewer reliability, Cohen’s Kappa was calculated [29] (Kappa = 0.899). To assess the risk of bias, Egger’s publication bias test [26] and Rosenthal’s failsafe-N [27] were calculated, and funnel plots were created. The ability of Egger’s test to detect bias when a meta-analysis is based on a small number of studies is limited [30]. In addition, the inclusion of at least 10 studies in the meta-analysis are necessary to perform it [26].

### 2.6. Statistical Analysis

R software, version 3.6.0., Copyright (C) 2019 (R Foundation for Statistical Computing, Vienna, Austria), was used with the metacont package to perform the meta-analysis. The forest plots were created using the forestplot package. For continuous data, the changes in the mean (M) and standard deviation (SD) between the baseline and final (pre–post-intervention) measurements of SBP and DBP values were used. Some studies had more than one experimental group and were treated as other subgroups in the analysis. We used the DerSimonian–Laird (Cohen’s) clustering method and assessed heterogeneity using Cochrane’s Q test (Chi^2^), Higgins’ I^2^, and significance (p) to determine the appropriateness of applying a fixed or random effects model for the pooled analysis. If there was evidence of between-study heterogeneity (I^2^ > 50%, *p* > 0.05), random-effect estimates were described [31].

To infer the pooled estimated standardised mean difference (MD), a meta-analysis with a random effects model was carried out [32]. The DerSimonian–Laird (Cohen’s) MD was interpreted by Cohen as small (0 to 0.2), medium (0.3 to 0.7), or large (≥0.8) [33]. Significant differences were determined at *p* < 0.05. 

## 3. Results

### 3.1. Search Results and Study Characteristics

The mean age of the sample was 60.7 ± 10.4 years, the mean duration of the intervention was 12 ± 8.3 weeks (range 4–32), and all interventions were supervised (Table 2).

### 3.2. Risk of Bias

Table 3 shows the score obtained on the PEDro scale for each of the articles included. The results show that quality obtained between eight and 10 points and was thus considered to be “high quality”. Figure 2 shows the risk of bias summary: the authors’ judgements are broken down according to each risk of bias criterion across all included studies.

### 3.3. Effect of the Interventions

All the included studies showed significant improvements in SBP and DBP levels [15,16,17,18,19,20,21,23] except for the study by Schroeder et al. [22], which showed no improvements in peripheral and central systolic blood pressure levels; however, it did show improvements in peripheral and central diastolic blood pressure levels. Seven studies [16,17,18,19,20,21,23] showed significant reductions (*p* < 0.05) in SBP and DBP levels in participants from the experimental group when compared to the control group; two of the studies [15,22] did not show much improvement when compared to the control group.

The meta-analysis showed a significant reduction in SBP (MD = −10.40, 95% CI −17.49 to −3.32, *p* < 0.001) and DBP (MD= −5.97, 95% CI −9.20 to −2.74, *p* < 0.001) levels (Figure 3 and Figure 4).

The meta-analysis for SBP (Table 4) and DBP (Table 5) indicate that multicomponent training showed improvements independent of age and the weekly frequency and intensity of aerobic training. All program durations showed significant improvements in DBP and SBP; however, those with a duration longer than 14 weeks produced significantly greater improvements (SBP: MD = 0.36, *p* = 0.043, DBP: MD = 0.40, *p* = 0.024). 

Similar results were observed for the duration of the training session in relation to SBP, with the most significant improvements observed for a duration of 30 min or longer. In relation to DBP, both protocols showed significant reductions, with no differences between them.

In the strength component, a work intensity of less than 75% of one repetition maximum (1 RM) (or equivalent) did not produce significant improvements in SBP though higher intensities did, with a significant difference observed between the two intensities used (*p* = 0.013). With respect to DBP, an intensity lower or higher than 75% of 1 RM produced significant improvements with a tendency towards significance, with no differences between the two. Any number of repetitions produced significant improvements in both SBP and DBP levels. However, a lower number of repetitions (≤10) resulted in significantly greater improvements in SBP than a higher number of repetitions (>10) (*p* = 0.005).

The programmes that used aerobic work first and strength work second did not show a significant improvement in SBP values; those that applied the components in reverse did show a significant reduction (MD = −2.65; *p* = 0.007); however, the difference between groups was not significant. For DBP, significant improvements with no differences between them were found for both cases.

## 4. Discussion

The main objective of the present study was to determine the effects of multicomponent training programmes on the SBP and DBP levels of adults with hypertension. The results confirm the first hypothesis formulated: multicomponent training will be effective in reducing systolic and diastolic hypertension levels in subjects with hypertension. The results showed that a multicomponent training programme achieved significant reductions in SBP and DBP levels in adults with hypertension, with these changes being significantly different from a control group. Individually, the works by Masroor et al. [21], dos Santos et al. [17], de Oliveira et al. [23], Guimaraes et al. [18], Sousa et al. [16], and Lima et al. [20] showed significant reductions in both SBP and DBP values post intervention with multicomponent training programmes. The study by Dos Santos et al. [17] showed the greatest effects on SBP and DBP levels, and the study by Schroeder et al. [22] only showed improvements in peripheral and central SBP levels and not in peripheral and central DBP levels. This difference may be because the latter study included only subjects with uncontrolled mild hypertension compared to the inclusion of subjects with monitored grades one and two hypertension in the other studies.

Our findings are in line with those from other systematic reviews and meta-analyses [12,13,14,34]. The works by Corso et al. [12], Herrod et al. [13], and Pescatello et al. [14] described significant reductions in SBP and DBP levels with multicomponent training. It should be noted that in the study by Herrod et al. [13], the intervention did not provide benefits that were any greater than single-mode training. In the review by Cornelissen et al. [34], only significant reductions in DBP levels were obtained with multicomponent training, without changes observed in SBP values. On the other hand, one of the most recent reviews, written by Saco-Ledo et al. [35], did not describe significant benefits on ambulatory blood pressure levels, specifying that significant benefits were only obtained with aerobic training programmes. However, the authors only included three studies in their systematic review that used multicomponent training, and one of them worked with people with mild hypertension, so the result may not be significant. The authors themselves indicated that more studies were needed for considering their results.

In this sense, multicomponent training as a physical activity programme is effective in playing an important role not only in primary but also in secondary cardiovascular prevention. In this regard, the effects of physical activity programmes on the metabolic environment and systemic chronic inflammation, as well as adaptations at the vascular and cardiac tissue levels, have been described [36,37]. As in the studies previously mentioned, those included in the present systematic review and meta-analysis were highly heterogeneous in terms of the design of the multicomponent training programmes, suggesting that this may have an influence on the outcome of the SBP and DBP responses [38,39], which will be discussed below.

These reductions in SBP and DBP can also be affected by drug treatment. In addition to commencing drug treatment, the patient with hypertension should make lifestyle changes, such as including regular physical exercise [40,41]. The studies by Sousa et al. [16], dos Santos et al. [17], Guimaraes et al. [18], Lima et al. [20], Masroor et al. [21], and de Oliveira et al. [23] included adults with monitored hypertension as the inclusion criteria. The participants in both the experimental and control groups had some pharmacological treatment that they continued to maintain during the study. After the completion of the multicomponent training interventions, the studies showed significant reductions in SBP and DBP levels, and these reductions differed from changes in the control group, who did not take part in the training programmes. However, the participants in the studies by Stewart et al. [15] and Schroeder et al. [22] had mild hypertension without pharmacological treatment as an inclusion criterion. The first [15] showed significant reductions in SBP and DBP; however, these did not differ from changes in the control group, and the second [22] showed a reduction in SBP, which was significantly different compared to the control group but not in DBP. This could be due to the fact that multicomponent training may be more effective for higher levels of hypertension, and the differential results could also be due to the differences in the programmes implemented, as will be discussed below.

The second objective of the present study was to identify the optimal dosage of multicomponent training for managing hypertension in adults. The results confirm the second hypothesis: the multicomponent training protocol that will show the greatest benefits for blood pressure will be one that applies moderate to vigorous intensities in cardiovascular work and moderate to high intensities in strength training. Age is one of the risk factors that influences blood pressure levels, with the risk of hypertension increasing from the ages of 60–65 years old [42]. The present meta-analysis shows that multicomponent training produced significant improvements in SBP and DBP regardless of age when comparing studies that included a sample younger than 65 years old [17,18,21,22,23] with studies that included a sample older than 65 [16,20]. These positive results are due to the fact that regular physical exercise can prevent or reverse pathologies associated with aging, as is the case with hypertension and as reflected in studies such as the one by Marcos-Pardo [9] in its multi-domain intervention programme Healthy-Age. 

Regarding the duration of the interventions, the results of the meta-analysis showed significant reductions in SBP and DBP levels with multicomponent training regardless of the duration of the programme. Of note, interventions lasting longer or equal to 14 weeks resulted in greater reductions in blood pressure levels. Acute adaptations to exercise occur from the start, with reductions of 10–20 mm Hg in SBP after the end of the training session, with these effects lasting up to 22 h. If exercise is maintained regularly over time, these adaptations will become increasingly larger and chronic, with reductions of up to 10 mm Hg in SBP and 8 mm Hg in DBP [43]. Training programmes aimed at improving and managing hypertension should be considered as lifelong strategies to maintain these adaptations. 

Regarding the frequency of the multicomponent training programmes, no significant differences were observed. The result of the meta-analysis indicated that reductions in SBP and DBP improved independently of the number of training days, with three days/week being the minimum frequency applied in the included studies. The exercise recommendations for hypertension from the American College of Sports Medicine and the latest hypertension clinical guidelines indicate the same frequency as the results obtained in this meta-analysis, with 3 days/week being the minimum training frequency recommended [3,44].

In relation to aerobic training, all the groups obtained significant reductions in SBP and DBP irrespective of work volume or intensity. However, the protocols that applied ≥30 min of work were shown to be more effective in reducing SBP. Regarding the minimum aerobic exercise time for improving hypertension, the main cardiology clinical guidelines state a minimum of 30 min per day, a length of time associated with reductions of 7 mm Hg for SBP and 5 mm Hg for DBP, with these guidelines in line with the results obtained in this meta-analysis.

In the protocols utilized for strength work, the studies that used a lower number of repetitions also showed a higher percentage of repetition maximum. The meta-analysis revealed that although both groups showed improvements, performing less than or equal to 10 repetitions produced greater reductions in SBP. In relation to the percentage of 1 RM, only work higher than 75% of 1 RM produced improvements in this variable. The DBP response was not different between protocols that applied intensities lower than 75% of 1 RM than higher ones, nor between protocols that performed more than 10 repetitions versus those that performed fewer repetitions. In this sense, it follows that a higher-volume and lower-intensity strength work protocol could have a greater influence on blood pressure in subjects with hypertension. Training programmes aimed at improving strength at low–moderate work intensities provide many benefits for the adult population, with this being an effective stimulus for improving blood pressure levels [45]. The exercise recommendations for subjects with hypertension refer to the practice of 150–300 min of aerobic exercise per week at a moderate–vigorous intensity, including 2–3 days of strength exercise per week, without specifying intensities, series, or repetitions, making it difficult to compare the findings from this meta-analysis [3].

The present meta-analysis did not show significant differences with respect to the order of the exercise components, although the changes in the order were only significant for the group that first performed strength exercises and then aerobics [17,18,20]. Some studies, such as those by Eklund et al. [46], Schumann et al. [47], or Wilhelm et al. [48], can explain these results. In their work, with different types of populations, sedentary women, sedentary older adults, and active young men, it was observed that the order of the components in multicomponent training programmes did not affect the adaptations of each modality.

Strengths and limitations:

The main limitation of this systematic review and meta-analysis is the small number of RCTs that currently exist, the heterogeneity of the population included (sex and age), the heterogeneity of the protocols, the types of aerobic and strength training exercises, and the small sample sizes in some studies. These aspects limit the comparisons between the different intervention groups and indicate the need for new studies to provide new data that allows for the generalisation of results.

However, the studies analysed herein also had strengths, among which we must underline that they obtained scores between 8 and 10 on the PEDRO scale, a ranking considered “high quality” [28].

## 5. Conclusions

The results of this study indicate that multicomponent training programmes significantly improved the SBP and DBP levels of adults with hypertension. The greatest benefits were found in programmes lasting longer than 14 weeks with a frequency of at least three sessions per week and sessions lasting approximately 60 min. The intensity of aerobic training should be around 75% of the reserve HR, and the intensity of strength training should be greater than 75% of 1 RM, with 10 repetitions or less per set. This research provides relevant information for professionals who prescribe physical activity to subjects with hypertension, providing key guidelines for its implementation. In this regard, there are several future areas for research in this field. It is necessary to investigate if there are differences in the results of the application of two protocols with training that apply the different components in different sessions or in the same session. It is also considered relevant to investigate the influence of the type of activity during cardiovascular work.

## Figures and Tables

**Figure 1 sports-11-00115-f001:**
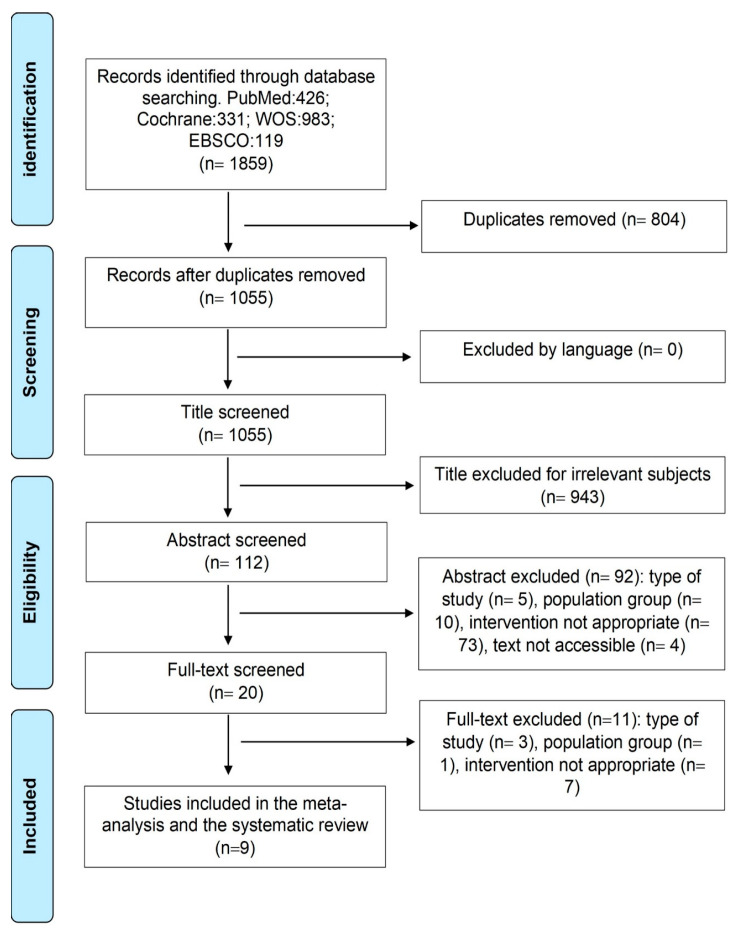
Flow chart of studies searched, selected and included.

**Figure 2 sports-11-00115-f002:**
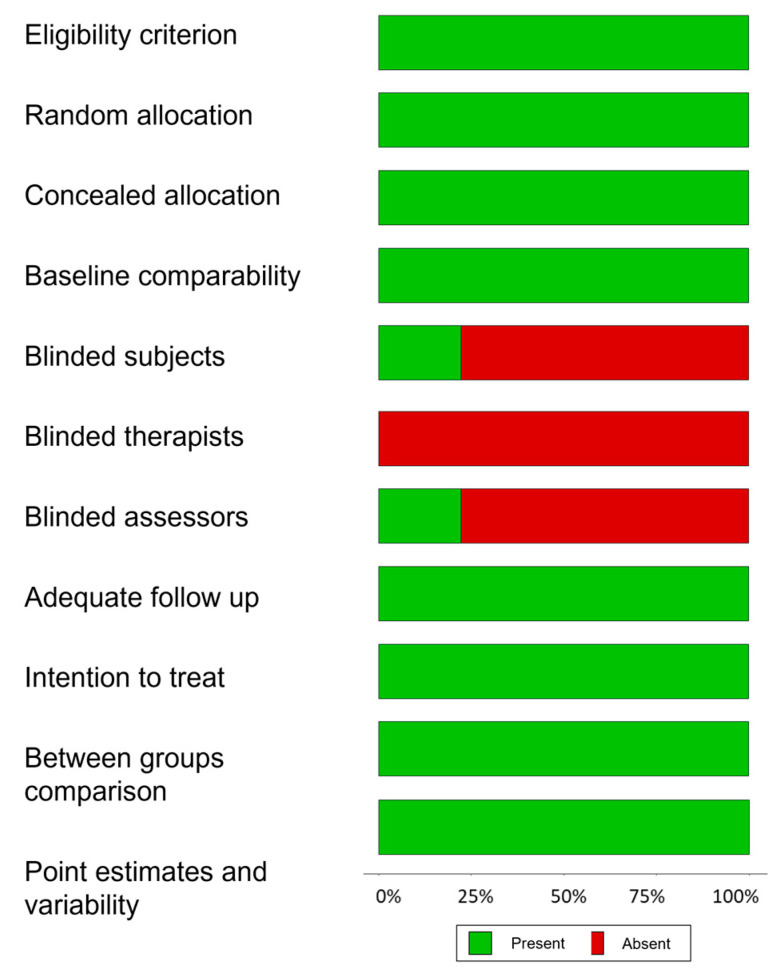
Risk of bias summary: authors’ judgements broken down according to each risk of bias criterion across all included studies.

**Figure 3 sports-11-00115-f003:**
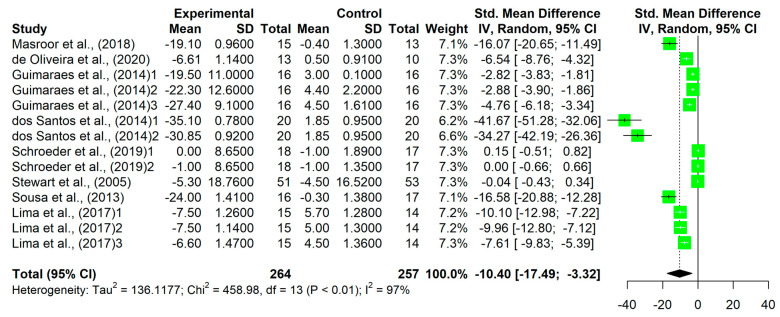
Forest plot comparing multicomponent training with supervision on changes in systolic blood pressure [15,16,17,18,20,21,22,23].

**Figure 4 sports-11-00115-f004:**
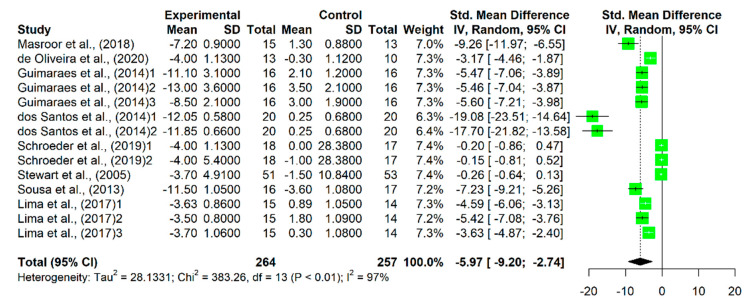
Forest plot comparing multicomponent training with supervision on changes in diastolic blood pressure [15,16,17,18,20,21,22,23].

**Table 1 sports-11-00115-t001:** Data extraction for each included study.

Studies	Sample	Age	Treatment	Inclusion Criteria	Main Variables
Oliveira et al. [23]	EG = 13 CG = 10	62.65 ± 6.0 years	Diuretics; ARBs II; ACE inhibitors	Age ≥ 50 years; controlled HTN; medical clearance for exercise; no physical activity in the last 6 months; no serious medical conditions	BP; isometric maximal strength; body composition; VO^2^
Schoeder et al. [22]	EG (MCT) = 18; (AT), (ST) and (CG) = 17	45–74 years 58 ± 7	Untreated	SBP 120–149 and DBP 80–99 mm Hg without medication; BMI 25–40 kg/m^2^; sedentary; no serious medical conditions; non-smoker; non-pregnant	BP; HR; BMI; body composition; cardiorespiratory fitness; strength
Masroor et al. [21]	EG = 15 CG = 13	30–50 years 40.45 ± 4.2	A drug (not specified)	Sedentary women; premenopausal; stage 1 or 2 HTN; no serious medical conditions; no physical activity in the last 6 months	BP; HRV
Lima et al. [20]	EG (MCT) and (AT)= 15; CG = 14	60–75 years	Hydrochlorothiazide; ACE inhibitors; ARBs II	Antihypertensive medication; SBP < 160 mm Hg and DBP 105 mm Hg; non-smokers; no serious medical conditions; not obese II or III	BP; blood markers; cardiac hypertrophy; body composition; VO^2^max
Son et al. [19]	EG = 10; CG = 10	72–85 years 75 ± 2	Untreated	Postmenopausal women; stage 1 HTN; non-obese; non-smokers; sedentary; non-medicated	BP; body composition; functional capacity; VO^2^max; arterial stiffness; endothelin-1; nitrate
Guimaraes et al. [18]	EG = 16; CG = 16	40–65 years 53.7 ± 6.0	Diuretics; CCBs; ACE inhibitors; ARBs II; Beta-blockers	HBP >5 years; use of three antihypertensive drugs; no serious medical conditions; no physical activity in the last 6 months; non-smokers	BP; VO^2^max
Dos Santos et al. [17]	EG (TST) = 20; (EST) and CG = 20	60–65 years 63.1 ± 2.3	CCBs; ACE inhibitors; ARBs II	Female; HTN; SBP < 180 and DBP < 110 mm Hg; sedentary; not diabetic; not on drugs and/or alcohol	BP; biochemical parameters
Sousa et al. [16]	EG (MCT) = 16 (AT) = 15, CG = 17	65–75 years (69.1 ± 5.0)	A drug (not specified)	Older men; no diabetes; no severe obesity; no severe hypertension; no neurological; mental or cognitive impairment; no physical impairment	BP; strength; lower limb aerobic endurance; body fatness
Stewart et al. [15]	EG = 51; CG = 53	55–75 years 63.6 ± 5.7	Untreated	Untreated mild hypertension; no physical activity in the last 6 months; non-smokers; no diabetes; no serious diseases	BP; VO^2^ max; strength; body composition; arterial stiffness

AT: aerobic training; ST: strength training; EST: eccentric strength training; TST: traditional strength training; MCT: multicomponent training; HR: heart rate; CG: control group; EG: experimental group.; HTN: arterial hypertension; BMI: body mass index; BP: blood pressure; DBP: diastolic blood pressure; SBP: systolic blood pressure; HRV: heart rate variability.

**Table 2 sports-11-00115-t002:** Intervention programmes of the studies analysed.

Studies	Start	Training	Finish	T, F, D
Oliveira et al. [23]	10′ running and S	Aerobic: 25′ on treadmill, progressing the intensity by 5% every two weeks from 70% HRR to 80% HRR. Strength: 6 exercises (rowing, standing bench press, arm curl, knee extension, knee flexion, forward march with a resistance band) × 2 sets × 15 repetitions × 30″ between sets and exercises. Intensity measured with OMNI-Resistance exercise scale (RES), progressing from 5 RPE progressing to 7 RPE and with the elastic bands from low extensions to high tensions.	10′ S and ME	8 wks, 3 days/wk, 70′
Schoeder et al. [22]	-	Aerobic: 30′ on treadmill or cycle ergometer at 40% HRR, progressing to 70% HRR. Strength: 30′ with 8 exercises (chest press, shoulder press, pull-down, lumbar extension, abdominal crunches, torso rotation, biceps curl, triceps extension, leg press, quadriceps extension, leg flexion and hip abduction) × 2 sets of 18–20 reps max until progressing to 3 sets of 10–14 reps max, with 1–2′ rest between sets.	-	8 wks, 3 days/wk, 60′
Masroor et al. [21]	5′ treadmill at 40% of HRmax	Aerobic: 20′ treadmill at 50–80% HRmax (intensity increased gradually over 4 weeks) Strength: 5 exercises (bicep curls, triceps extensions, abdominal crunches, leg curls, and knee extensions) × 3 sets × 10 reps at an intensity of 50–80% of 1 RM (intensity was gradually increased over 4 weeks).	5′ treadmill at 40% of HRmax	4 wks, 5 days/wk, 50′–60′
Lima et al. [20]	5′	Strength: 9 strength exercises (leg press, 45° leg press or bench press, bench extensor, front bench with handle, seated bench press, upright rowing, plantar flexion, seated rowing and crunches) × 15 repetitions (upper limbs) and 20 repetitions (lower limbs and trunk) with an intensity of 50–60% 1 RM. From week 1 to 4, they performed 1 round, and from week 5 to 10, 2 rounds. Aerobic: 20′ on treadmill from week 1 to 4 and 30′ from week 5 to 10. The intensity of the exercise was based on the physical condition of each participant.	-	10 wks, 3 days/wk, 40′–60′
Son et al. [19]	5′ SS	Strength: 20′ exercises with elastic bands; 10 exercises (seated row, bicep curl, shoulder flexion, elbow flexion, push-up; hip flexion, hip extension, calf raise, leg press, and squats). Aerobic: 30′ of walking at an exercise intensity of 40% to 50% HRR from weeks 1 to 4 and 60% to 70% HRR from weeks 9–12.	5′ SS	12 wks, 3 days/wk 60′
Guimaraes et al. [18]	5′	Strength: 20′ calisthenics in water (upper and lower limbs). Aerobic: 30′ walking in water between 11 and 13 on the Borg scale.	5′ S	12 wks, 3 days/wk, 60′
Dos Santos et al. [17]	TST	Strength: 7 exercises (bench press with barbell, leg press, trunk extension, leg extension, arm curl, dorsiflexion, and lateral raises). Week 1 to 5 at 70% 10 RM, week 6 to 11 at 80% of 10 RM, and week 12 to 16 at 90% of 10 RM. 3 sets × 10 reps. Aerobic: 20′ on a treadmill at 65–75% HRR.	-	16 wks, 3 days/wk, 50′–60′
	EST	Strength: work 30% 1 RM more than in EFT. Same resistance work.	-	
Sousa et al. [16]	10′ walking and S	Two days of training on land and one day in the water. Aerobic: 30′ with a choice between walking, jogging, or dancing with moderate intensity; in addition, 10′ of muscle resistance with 3 exercises (with own body weight, and upper and lower body), 3 sets, 15–20 repetitions. The aquatic session: relay races, water volleyball, and water polo. Strength: 6 exercises (bench press, leg press, lateral leg extension, leg extension, leg curl, military press, leg curl, and arm curl) at 65%R 1 RM × 3 sets × 10–12 reps (MC1); 75%RM. 24 × 3 sets × 8–10 reps (MC2); 70%RM × 3 sets × 8–10 reps (MC3); 65%RM × 3 sets × 10–12 reps (MC4).	5′ S	32 wks, 3 days/wk, 60′
Stewart et al. [15]	S	Strength: 7 exercises (latissimus dorsi pull down, leg extension, leg curl, bench press, leg press, shoulder press, and seated mid-row) × 2 sets × 10–15 reps at 50% 1 RM. Aerobic: 45′ with a choice of treadmill, exercise bike, or stair climber. Intensity between 60 and 90% HRmax.	-	24 wks, 3 days/wk, >60′

′: minutes; 1 RM: 1 repetition maximum; D: duration/session; S: stretching; SS: static stretching; EST: eccentric strength training; TST: traditional strength training; F: weekly frequency; HRmax: maximum heart rate; HRR: heart rate reserve; MC: microcycle; ME: mobility exercise; Reps: repetitions; WKS: weeks; T: intervention time; RPE: rate of perceived exertion.

**Table 3 sports-11-00115-t003:** Assessment of the methodological quality (PEDro scale) of the articles included.

	C1	C2	C3	C4	C5	C6	C7	C8	C9	C10	C11	Total
Oliveira et al. [23]	1	1	1	1	1	0	0	1	1	1	1	9
Schroeder et al. [22]	1	1	1	1	0	0	1	1	1	1	1	9
Masroor et al. [21]	1	1	1	1	1	0	1	1	1	1	1	10
Lima et al. [20]	1	1	1	1	0	0	0	1	1	1	1	8
Son et al. [19]	1	1	1	1	0	0	0	1	1	1	1	8
Guimaraes et al. [18]	1	1	1	1	0	0	0	1	1	1	1	8
Dos Santos et al. [17]	1	1	1	1	0	0	0	1	1	1	1	8
Sousa et al. [16]	1	1	1	1	0	0	0	1	1	1	1	8
Stewart et al. [15]	1	1	1	1	0	0	0	1	1	1	1	8

C1: choice criteria were specified; C2: subjects were randomly assigned to groups; C3: assignment was concealed; C4: groups were similar at baseline in relation to the most important prognostic indicators; C5: all subjects were blinded; C6: all therapists administering therapy were blinded; C7: all assessors measuring at least one key outcome were blinded; C8: measures of at least one of the key outcomes were obtained from more than 85% of the subjects initially assigned to the groups; C9: results were presented for all subjects who received treatment or were assigned to the control group or, when this could not be carried out, data for at least one key outcome were analysed on an intention-to-treat basis; C10: results of statistical comparisons between groups were reported for at least one key outcome; C11: the study provides point-in-time and variability measures for at least one key outcome.

**Table 4 sports-11-00115-t004:** Analysis of the effect of multicomponent training on SBP in patients with hypertension according to duration in weeks, weekly frequency, type of physical exercise programme applied, intensity, volume, and order of components.

	Authors	G	MD	95% CI	*p*	MD	95% CI	*p*
Age								
<65 years	Masroor et al. [21], de Oliveira et al. [23], Guimaraes et al. [18], dos Santos et al. [17]_1_TRT, dos Santos et al. [17]_2_ERT, and Schroeder et al. [22]	6	−2.2	−3.63; −0.78	0.002	−0.17	−0.58; 0.23	0.400
≥65 years	Sousa et al. [16] and Lima et al. [20]	2	−1.06	−1.93; 0.2	0.016			
Weeks								
<14 weeks	Masroor et al. [21], de Oliveira et al. [23], Guimaraes et al. [18], Schroeder et al. [22], and Lima et al. [20]	5	−0.97	−1.74; 0.21	0.012	0.36	−0.01; 0.70	0.043
≥14 weeks	dos Santos et al. [17]_1_TRT, dos Santos et al. [17]_2_ERT, and Sousa et al. [16]	3	−3.51	−5.74; −1.27	0.002			
Frequency								
3 days/weeks	de Oliveira et al. [23], Guimaraes et al. [18], dos Santos et al. [17], Schroeder et al. [22], Sousa et al. [16], and Lima et al. [20]	6	−1.47	−2.53; −0.40	0.006	0.42	0.97; −0.12	0.130
>3 days/weeks	Masroor et al. [21]	1	−2.63	−3.63; −1.64	˂0.001			
Intensity Aerobic Training								
≤75% HRR	Masroor et al. [21], Guimaraes et al. [18], dos Santos et al. [17]_1_TRT, dos Santos et al. [17]_2_ERT, and Schroeder et al. [22]	5	−1.51	−4.26; −0.77	0.004	−0.05	−0.47; 0.37	0.819
>75% HRR	de Oliveira et al. [23] and Sousa et al. [16]	2	−1.15	−1.87; −0.43	0.001			
Duration Aerobic Training								
<30 min	de Oliveira et al. [23], dos Santos et al. [17]_1_TRT, dos Santos et al. [17]_2_ERT, and Lima et al. [20]	4	−2.58	−4.61; −0.55	0.012	0.46	0.16; 0.76	0.002
≥30 min	Masroor et al. [21], Guimaraes et al. [18], Schroeder et al. [22], Stewart et al. [15], and Sousa et al. [16]	5	−5.25	−8.14; 2.35;	<0.001			
Intensity Strength Training								
≤75% 1 RM	Schroeder et al. [22], Sousa et al. [16], and Lima et al. [20]	3	−0.69	−1.55; 0.16	0.112	0.56	0.12;1	0.013
>75% 1 RM	Masroor et al. [21] and dos Santos et al. [17]_1_TRT	2	−4.03	−6.86; −1.21	0.005			
Repetitions Strength Training								
≤10	Masroor et al. [21] and dos Santos et al. [17]_1 TRT	2	−4.03	−6.86; −1.21	0.005	−0.61	−1.03; −0.18	0.005
>10	de Oliveira et al. [23], Schroeder et al. [22], Sousa et al. [16], and Lima et al. [20]	4	−0.71	−1.33; −0.08	0.026			
Order Training Components								
Aerobic + Strength	Masroor et al. [21], de Oliveira et al. [23], and Schroeder et al. [22]	3	−1.10	−2.52; 0.33	0.131	0.22	−0.16; 0.59	0.256
Strength + Aerobic	Guimaraes et al. [18], dos Santos et al. [17]_1_TRT, dos Santos et al. [17]_2_ERT, and Lima et al. [20]	4	−2.65	−4.59; −0.71	0.007			

MD: mean difference; HRR = heart rate reserve; CI: confidence interval; *p*: significance; MR: maximum repetition.

**Table 5 sports-11-00115-t005:** Analysis of the effect of multicomponent training on DBP in patients with hypertension according to duration in weeks, weekly frequency, type of physical exercise programme applied, intensity, volume, and order of components.

	Authors	G	MD	95% CI	*p*	MD	95% CI	*p*
Age								
<65 years	Masroor et al. [21], de Oliveira et al. [23], Guimaraes et al. [18], dos Santos et al. [17]_1_TRT, dos Santos et al. [17]_2_ERT, and Schroeder et al. [22]	6	−1.79	−2.67; −0.91	<0.001	−0.20	−0.60; 0.2	0.326
≥65 years	Sousa et al. [16] and Lima et al. [20]	2	−0.97	−1.60; −0.34	0.002			
Weeks								
<14 weeks	Masroor et al. [21], de Oliveira et al. [23], Guimaraes et al. [18], Schroeder et al. [22], and Lima et al. [20]	5	−1.04	−1.55; 0.54	<0.001	0.40	0.75; 0.05	0.024
≥14 weeks	dos Santos et al. [17] _1_TRT, dos Santos et al. [17] _2_ERT, and Sousa et al. [16]	3	−2.51	−3.83; −1.18	<0.001			
Frequency								
3 days/weeks	de Oliveira et al. [23], Guimaraes et al. [18], dos Santos et al. [17] _1_TRT, Schroeder et al. [22], Sousa et al. [16], and Lima et al. [20]	6	−1.46	−2.26; 0.66	<0.001	−0.08	0.47; −0.62	0.784
>3 days/weeks	Masroor et al. [21]	1	−1.17	−1.95; 0.39	0.003			
Intensity Aerobic training								
≤75% HRR	Masroor et al. [21], de Oliveira et al. [23], Guimaraes et al. [18], dos Santos et al. [17] _1_TRT, dos Santos et al. [17] _2_ERT, and Schroeder et al. [22]	6	1.79	−2.67; 0.91	<0.001	−0.06	−0.58; 0.47	0.832
>75% HRR	Sousa et al. [16]	1	−1.3	−2.07; −0.53	<0.001			
Intensity Strength Training								
≤75% 1 RM	Schroeder et al. [22], Sousa et al. [16], and Lima et al. [20]	3	−1.3	−2.06; −0.55	<0.001	0.21	0.64; −0.23	0.346
>75% 1 RM	Masroor et al. [21] and dos Santos et al. [17]_1_TRT	2	−2.35	−4.71; 0.01	0.050			
Duration Aerobic training								
<30 min	de Oliveira et al. [23], dos Santos et al. [17]_1_TRT, dos Santos et al. [17] _2_ERT, and Lima et al. [20]	4	−1.84	−3.28; −0.39	0.012	0.21	−0.09; 0.51;	0.173
≥30 min	Masroor et al. [21], Guimaraes et al. [18], Schroeder et al. [22], Stewart et al. [15], and Sousa et al. [16]	5	−3.05	−5.26; −0.85	0.006			
Repetitions Strength Training								
≤10	Masroor et al. [21] and dos Santos et al. [17]_1_TRT	2	−2.35	−4.71; 0.01	0.050	−0.26	0.15; −0.68	0.216
>10	de Oliveira et al. [23], Schroeder et al. [22], Sousa et al. [16], and Lima et al. [20]	4	−1.10	−1.76; 0.44	0.001			
Order Training Components								
Aerobic + Strength	Masroor et al. [21], de Oliveira et al. [23], and Schroeder et al. [22]	3	−1.21	−2.07; −0.36	0.005	0.16	−0.21; 0.53	0.398
Strength + Aerobic	Guimaraes et al. [18], dos Santos et al. [17] _1_TRT, dos Santos et al. [17] _2_ERT, and Lima et al. [20]	4	−1.95	−3.27; −0.64	0.003			

MD: mean difference; HRR = heart rate reserve; CI: confidence interval; *p*: significance; MR = maximum repetition.

## Data Availability

No new data were created or analyzed in this study. Data sharing is not applicable to this article.

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
