# Peer review of "Multicomponent Training and Optimal Dosing Strategies for Adults with Hypertension: A Systematic Review and Meta-Analysis of Randomized Controlled Trials"

_sports, 2023, doi:10.3390/sports11060115_

Round 1
Reviewer 1 Report
General comments
The authors have clearly stated that the purpose of the study was to assess the impact of multicomponent training on the blood pressure of hypertensive adults and ascertain the nature of the dose-response relationship. The paper is well-written, easy to follow and adds merit to the vital role of integrated exercise training programs in the management of hypertension. Given this approach, this work can enhance future attempts in similar research area. However, I have highlighted a few minor suggestions and concerns in my specific comments section (below) that need to be addressed before accepting this work for publication. In summary, this is an innovative and extensive work that might be considered for publication after revising the initial manuscript as needed.
Specific comments
THROUGOUT MANUSCRIPT
- Please use people-first language by changing the term “hypertensive adults/individuals/patients” to “adults/individuals/patients with hypertension”, aiming to avoid disease stigma.
INTRODUCTION
- I suggest adding a sentence about the beneficial role of multicomponent exercise programs in health according to the latest guidelines by the World Health Organization (1).
- A statement about the popularity of exercise for populations struggling with most common health diseases in the health and fitness industry worldwide but also in Spain according to the latest report of the American College of Sports Medicine (2), it could be a useful addition.
Suggested References:
1. https://pubmed.ncbi.nlm.nih.gov/33239350/
2. https://journals.lww.com/acsm-healthfitness/Fulltext/2023/01000/2023_Fitness_Trends_from_Around_the_Globe.7.aspx
DISCUSSION
- Reference 36 should be replaced by https://pubmed.ncbi.nlm.nih.gov/15076798/
- Discuss the results observed for various exercise interventions (4-6) as a critical piece of behavioral treatments for people living with cardiometabolic health complications.
- Strengths and limitations should be presented in a separate paragraph at the end of the discussion section. A heading is also needed.
- In conclusions, you should suggest future research attempts in this area while highlighting potential practical implications in a clearer way.
Minor editing of English language is required.
Author Response
Letter to reviewer reviewed 1
General comments
The authors have clearly stated that the purpose of the study was to assess the impact of multicomponent training on the blood pressure of hypertensive adults and ascertain the nature of the dose-response relationship. The paper is well-written, easy to follow and adds merit to the vital role of integrated exercise training programs in the management of hypertension. Given this approach, this work can enhance future attempts in similar research area. However, I have highlighted a few minor suggestions and concerns in my specific comments section (below) that need to be addressed before accepting this work for publication. In summary, this is an innovative and extensive work that might be considered for publication after revising the initial manuscript as needed.
- We appreciate your valuable comments and your time spent to improve our article.
Specific comments
THROUGOUT MANUSCRIPT
- Please use people-first language by changing the term “hypertensive adults/individuals/patients” to “adults/individuals/patients with hypertension”, aiming to avoid disease stigma.
- Thank you for the comment. We change the term.
INTRODUCTION
- I suggest adding a sentence about the beneficial role of multicomponent exercise programs in health according to the latest guidelines by the World Health Organization (1).
- Thank you for the comment. We have included it.
- A statement about the popularity of exercise for populations struggling with most common health diseases in the health and fitness industry worldwide but also in Spain according to the latest report of the American College of Sports Medicine (2), it could be a useful addition.
Suggested References:
- https://pubmed.ncbi.nlm.nih.gov/33239350/
- https://journals.lww.com/acsm-healthfitness/Fulltext/2023/01000/2023_Fitness_Trends_from_Around_the_Globe.7.aspx
- Thank you for the comment. We have included it.
DISCUSSION
- Reference 36 should be replaced by https://pubmed.ncbi.nlm.nih.gov/15076798/
- Thank you for the comment. This was replaced.
- Discuss the results observed for various exercise interventions (4-6) as a critical piece of behavioral treatments for people living with cardiometabolic health complications.
- Thank you for the comment. This was included.
- Strengths and limitations should be presented in a separate paragraph at the end of the discussion section. A heading is also needed.
- Thak you for the comment. This recommend has been done.
- In conclusions, you should suggest future research attempts in this area while highlighting potential practical implications in a clearer way.
- Thank you for the comment. This recommend has been included.

Reviewer 2 Report
Dear Editor,
I appreciate the opportunity to review the manuscript “Multicomponent Training and Optimal Dosing Strategies for Hypertensive Adults: A Systematic Review and Meta-Analysis of Randomized Controlled Trials”. The aim of the manuscript was to assess the impact of multicomponent training on the blood pressure of hypertensive adults and ascertain the nature of the dose-response relationship. The manuscript is interesting and adheres to the area of the Journal Sports (MDPI). However, it is not innovative and there are already systematic reviews with a similar approach to the one discussed in this manuscript. I also point out that the authors themselves cite some reviews similar to this one (Corso LM, Macdonald HV, Johnson BT, Farinatti P, Livingston J, Zaleski AL, et al. Is Concurrent Training Efficacious Antihypertensive Therapy? A Meta-analysis. Med Sci Sports Exerc. 2016;48(12):2398-406; Herrod PJJ, Doleman B, Blackwell JEM, O'Boyle F, Williams JP, Lund JN, et al. Exercise and other nonpharmacological strategies to reduce blood pressure in older adults: a systematic review and meta-analysis. J Am Soc Hypertens. 2018;12(4):248-67; Pescatello LS, Buchner DM, Jakicic JM, Powell KE, Kraus WE, Bloodgood B, et al. Physical Activity to Prevent and Treat Hypertension: A Systematic Review. Med Sci Sports Exerc. 2019;51(6):1314-23). Thus, I believe that the journal's editor should reflect on whether it is convenient to have this research published. With regard to the sections of this manuscript, I indicate some suggestions for improvement:
1 – The introduction is shortened. I suggest deepening the knowledge about the main content of this manuscript.
2 – In the introduction there are several statements without any reference author being indicated (Examples: The effect of cardiovascular training on blood pressure levels has been well established in the literature, with reductions ranging from -4.9 to -12 mm Hg for systolic blood pressure (SBP) and -3.4 to -5.8 mm Hg for diastolic blood pressure (DBP).; Cardiovascular disease is the leading cause of premature death worldwide, and hypertension is one of its most prevalent risk factors).
3 – From the theoretical framework, the authors must indicate the hypotheses for this research.
4 – In the methods section, show Figure 1. This figure is part of the methods and not the results.
5 – Carry out the discussion based on the hypotheses mentioned in the introduction.
In view of the above, I believe that the editor will need to decide whether he wants this manuscript to proceed with publication, given that there are recent systematic reviews on the same topic. If you move with the publication of the manuscript, I ask that the suggestions be accepted and that there are mandatory corrections.
Sincerely,
Reviewer.
Author Response
Letter to Reviewer 2
I appreciate the opportunity to review the manuscript “Multicomponent Training and Optimal Dosing Strategies for Hypertensive Adults: A Systematic Review and Meta-Analysis of Randomized Controlled Trials”. The aim of the manuscript was to assess the impact of multicomponent training on the blood pressure of hypertensive adults and ascertain the nature of the dose-response relationship. The manuscript is interesting and adheres to the area of the Journal Sports (MDPI). However, it is not innovative and there are already systematic reviews with a similar approach to the one discussed in this manuscript. I also point out that the authors themselves cite some reviews similar to this one (Corso LM, Macdonald HV, Johnson BT, Farinatti P, Livingston J, Zaleski AL, et al. Is Concurrent Training Efficacious Antihypertensive Therapy? A Meta-analysis. Med Sci Sports Exerc. 2016;48(12):2398-406; Herrod PJJ, Doleman B, Blackwell JEM, O'Boyle F, Williams JP, Lund JN, et al. Exercise and other nonpharmacological strategies to reduce blood pressure in older adults: a systematic review and meta-analysis. J Am Soc Hypertens. 2018;12(4):248-67; Pescatello LS, Buchner DM, Jakicic JM, Powell KE, Kraus WE, Bloodgood B, et al. Physical Activity to Prevent and Treat Hypertension: A Systematic Review. Med Sci Sports Exerc. 2019;51(6):1314-23). Thus, I believe that the journal's editor should reflect on whether it is convenient to have this research published. With regard to the sections of this manuscript, I indicate some suggestions for improvement:
- First of all we would like to thank the reviewer for his time and valuable comments in order to improve our manuscripts. Secondly, we have tried to clarify the difference between the present systematic review with meta-analysis and the previous ones discussed, thus justifying the need for the present meta-analysis.
1 – The introduction is shortened. I suggest deepening the knowledge about the main content of this manuscript.
- Thank you for the comment. We have extended the introduction.
2 – In the introduction there are several statements without any reference author being indicated (Examples: The effect of cardiovascular training on blood pressure levels has been well established in the literature, with reductions ranging from -4.9 to -12 mm Hg for systolic blood pressure (SBP) and -3.4 to -5.8 mm Hg for diastolic blood pressure (DBP).; Cardiovascular disease is the leading cause of premature death worldwide, and hypertension is one of its most prevalent risk factors).
- Thank you for the comment. We have included cites.
3 – From the theoretical framework, the authors must indicate the hypotheses for this research.
- Thank you for the comment Hypotheses have been included.
4 – In the methods section, show Figure 1. This figure is part of the methods and not the results.
- Thank you for the comment. The change has been done.
5 – Carry out the discussion based on the hypotheses mentioned in the introduction.
- Thank you for the comment. The change has been done.
In view of the above, I believe that the editor will need to decide whether he wants this manuscript to proceed with publication, given that there are recent systematic reviews on the same topic. If you move with the publication of the manuscript, I ask that the suggestions be accepted and that there are mandatory corrections.
- Thank you

Round 2
Reviewer 2 Report
Dear Editor,
thank you for the opportunity to review the manuscript again. In view of the changes made, the authors must to make the following changes:
1 - Put the hypotheses after the objectives
2 - On line 271, insert the term "In this..." The letter "i" is missing.
Thus, the manuscript should be accepted after these simple corrections. It is not necessary to carry out a new revision, in view of the simplicity of the changes.
Sincerely,
Reviewer